# Development of Plasma Protein Classification Models for Alzheimer’s Disease Using Multiple Machine Learning Approaches

**DOI:** 10.3390/ijms262311673

**Published:** 2025-12-02

**Authors:** Amy Tsurumi, Catherine M. Cahill, Andy J. Liu, Pranam Chatterjee, Sudeshna Das, Ami Kobayashi

**Affiliations:** 1Department of Surgery, Massachusetts General Hospital and Harvard Medical School, 55 Fruit St., Boston, MA 02114, USA; 2Neurochemistry Laboratory, Department of Psychiatry, Massachusetts General Hospital and Harvard Medical School, 149 13th St., Charlestown, MA 02129, USA; ccahill@helix.mgh.harvard.edu; 3Department of Neurology, Duke University School of Medicine, 3116 N. Duke St., Durham, NC 27704, USA; andy.liu@duke.edu; 4Duke-UNC Alzheimer’s Disease Research Center, 2424 Erwin Rd, Durham, NC 27705, USA; 5Department of Bioengineering, University of Pennsylvania, 210 South 33rd Street, Philadelphia, PA 19104, USA; pranam@seas.upenn.edu; 6Department of Neurology, Massachusetts General Hospital and Harvard Medical School, 65 Lansdowne St., Cambridge, MA 02139, USA; sdas5@mgh.harvard.edu; 7Department of Neurology, Brigham and Women’s Hospital and Harvard Medical School, 60 Fenwood Rd., Boston, MA 02115, USA; akobayashi1@bwh.harvard.edu

**Keywords:** Alzheimer’s disease, neurodegeneration, aging, machine learning, biomarkers, proteomics, diagnosis

## Abstract

Alzheimer’s Disease (AD) management is challenging due to limitations in detection methods. Currently, cerebrospinal fluid (CSF) biomarkers involve assessing β-amyloid (Aβ) and phosphorylated tau proteins. The lumbar puncture procedure to obtain CSF is invasive and sometimes causes significant anxiety in patients. In contrast, plasma biomarkers would allow rapid, accurate, and cost-effective diagnosis, while minimizing invasiveness and discomfort. Using a dataset involving 120 plasma proteins from clinically diagnosed AD patients versus cognitively normal subjects, we developed classification models by applying various machine learning algorithms (EBlasso, EBEN, XGBoost, LightGBM, TabNet, and TabPFN) to plasma proteomic measurements. Gene ontology and pathway enrichment, and a literature review were used to evaluate the potential relevance of the biomarkers identified in AD-related mechanisms. Biomarkers identified were also evaluated for the enrichment of aging-related biomarkers. The models developed yielded high AUROC and accuracy, mostly >0.9. Proteins selected as predictors by all the models included Angiopoietin-2 (ANG-2), epidermal growth factor (EGF), Interleukin 1α (IL-1α), and platelet growth factor subunit B (PDGF-BB). Ample previous literature supported their relevance in AD. The pool of all the biomarkers identified was significantly enriched with known aging-related biomarkers (*p* = 0.040). Applying cutting-edge algorithms is expected to be advantageous for developing AD prediction models with plasma proteomic data, and future large studies to externally validate the constructed models in other populations to assess their generalizability is important. The proteins uncovered may represent novel preventative or therapeutic targets.

## 1. Introduction

Alzheimer’s Disease (AD) cases in the US are currently estimated to exceed 6 million and are projected to increase to a staggering 13.8 million by 2060 [1]. AD is the most common basis for dementia and the sixth leading cause of death [1,2]. Additionally, it imposes an enormous economic burden on society, as AD-associated healthcare costs were estimated to be $321 billion in 2022, and are expected to surge to above $1 trillion by 2050 [1]. These figures highlight the need to devise new strategies for AD management as an urgent public health priority area.

One major challenge in AD management is the dearth of effective and rapid detection methods. Currently, positron emission tomography (PET) imaging and cerebrospinal fluid (CSF) biomarkers involve assessing β-amyloid (Aβ) and phosphorylated tau peptides to monitor and diagnose AD are used [3]. The lumbar puncture procedure for CSF collection is invasive, could cause significant anxiety, and could be dangerous to specific groups of patients, such as those with structural brain lesions or thrombocytopenia and other conditions necessitating anticoagulation treatment [4]. Moreover, PET imaging centers are costly, labor intensive, and more common at larger hospitals, decreasing access and widespread adoption. These limitations make the widespread adoption of PET imaging and CSF biomarkers difficult, underscoring the urgent need for novel biomarkers for diagnosing AD rapidly, accurately, and cost-effectively while reducing invasiveness and discomfort to patients.

Plasma, which can even be collected as part of routine laboratory work in various clinical settings, such as primary care or geriatrician offices, offers an attractive solution for minimally invasive and cost-effective biomarkers to facilitate the wide adoption of early screening and longitudinal monitoring. The measurement of various Aβ and phosphorylated tau peptides in plasma or serum, similar to those measured in CSF, or other markers of neurodegeneration, has been reported in a large number of studies [5,6,7,8,9,10,11,12,13,14,15,16,17,18,19]. Commonly tested peptides include Aβ_1–42_, phosphorylated-Tau (p-Tau) at various residues, and total-Tau (t-Tau). Recently, the Lumipulse G plasma p-Tau217/Aβ_1–42_ ratio blood test was shown to detect abnormal Aβ- and Tau-positron emission tomography (PET) with high accuracy [20] and became the first blood test for AD diagnosis to be approved by the U.S. Food and Drug Administration [21].

Other relatively recent studies have shown that brain-derived p-Tau more specifically correlates with Tau-PET and cognition compared to t-Tau [18], or that Tau microtubule-binding region (MTBR) containing the residue 243 (MTBR-tau243) can better detect insoluble Tau aggregates compared to common p-Tau measures later in more advanced stages of AD [22], suggesting that additional studies to determine which specific Aβ and Tau peptides can best detect AD pathology may also be beneficial. Additional challenges with the approaches based on plasma Aβ and Tau include challenges in establishing reference ranges and measures becoming elevated for reasons other than AD, including common comorbidities such as prior myocardial infarction, stroke, or chronic kidney disease [14,23,24]. Differences in race have also been reported [14], and, moreover, AD pathogenesis is considered to be a heterogeneous condition involving additional pathways other than the peptides previously known to be related to neurofibrillary tangles (NFTs) [25,26]. Consequently, additional studies to investigate novel plasma AD biomarkers more broadly may still be needed to improve the accuracy of detection and elucidate previously unknown molecular mechanisms, or to further understand the contribution of previously implicated mechanisms.

Various recent machine learning (ML) algorithms provide effective means for analyzing large molecular datasets, which have yet to be applied to AD plasma proteomic datasets. AD proteomic ML studies have mostly evaluated CSF [27,28,29,30,31] or brain [29,32,33,34], rather than plasma, which would not allow for clinically useful, non-invasive biomarkers. Previous AD plasma proteomic ML studies used predictive analysis of microarrays (PAM) [35], random forest [36], or support vector machine (SVM) [37,38,39], which are older algorithms that lack interpretability. A more recent study used the least absolute shrinkage and selection operator (Lasso) to develop accurate prediction models for the prognosis of MCI patients who progress to dementia [40], and another recent study applied Light Gradient Boosting Machine (LightGBM) to classify AD versus cognitively normal status [41]. These studies included evaluating both protein alone and protein with clinical information associated with AD and its pathological outcome (demographic and cognition). A recent study profiled thousands of plasma proteins from a large number of samples with the SomaScan 7k platform and also used Lasso to develop prediction models for AD clinical status as well as AD biomarker status [42]. Another recent study using a large, harmonized dataset derived from the SomaScan platform developed prediction models for APOE ε4 status more widely across AD, PD, FT and ALS, rather than developing prediction models for AD clinical outcome, and evaluated the correlation between different organ aging [43]. Other studies that developed AD classification models from plasma or serum protein measures evaluated a select panel of proteins previously known to be related to NFTs and inflammation rather than employing an unbiased approach to identify previously unknown AD-associated proteins [34,35,44,45,46], or by functional network analysis [47]. One study evaluating miRNAs [48] used random forest, and another study analyzed plasma metabolites with deep learning, random forest, and eXtreme Gradient Boosting (XGBoost) [49]. Other studies were designed to identify plasma proteins associated with AD rather than to develop classification models [50,51,52]. More recently, additional machine learning algorithms for tabular data have become available, and, therefore, testing cutting-edge algorithms is expected to further improve prediction modeling using plasma proteins. 

In this study, we tested various ML algorithms, most of which have not yet been tested in AD plasma proteomic data analysis but may be more advantageous. We first applied the fast empirical Bayesian Lasso (EBlasso) [53] and empirical Bayesian Elastic Net (EBEN) [54], described to improve analysis with abundant multicollinearity. We also tested XGBoost [55] and LightGBM [56] gradient-boosted decision tree (GBDT) methods, with SHapely Additive exPlanations (SHAP) [57,58] to allow for interpretable models. Furthermore, we applied TabNet [59] and Tabular Prior-Data Fitted Network (TabPFN) [60,61], representing more recent advancements in interpretable tabular deep learning. In particular, TabPFN is based on a pre-trained foundation model, which is a novel approach to model development. We analyzed the proteomic datasets provided as part of the Appendix A of a previous study [35], which was published many years before these methods were developed. Assessing whether the results of the prediction model developed using these newer algorithms over the original study’s model prediction results is expected to help understand whether advancements in ML prediction analysis may advance AD detection using plasma proteomics.

Given the advantage of developing interpretable models, we also aimed to evaluate pathways and previous reports of how the proteins identified may be mechanistically related to AD. Moreover, we evaluated whether the biomarkers found to be important for classifying AD are enriched with those shown to be related to aging by being identified as part of aging clock models, differentially regulated by age [62,63,64,65,66,67,68,69,70,71], or among those identified in the Human Ageing Genomic Resources (HAGR) database (GenAge, CellAge and cell senescence signatures) [72,73].

## 2. Results

### 2.1. Applying Various ML Algorithms to Previously Generated Plasma Proteomic Data Yielded Highly Accurate Classification Models

The plasma proteomic dataset provided as Appendix A by a previous study [35] included measures of 120 plasma proteins from confirmed Alzheimer’s Disease (AD) and cognitively normal (CN) subjects. There were 83 subjects included in the training set (AD, *n* = 43 and CN, *n* = 40) and 81 in the test set (AD, *n* = 42 and CN, *n* = 39). The original study performed predictive analysis of microarrays (PAM) to develop AD outcome classification models using the training set, then assessed their performance in the test set—this model included 18 proteins and provided 89% accuracy in both the training and test sets [35]. Our objective was to utilize the same dataset to test whether newer and interpretable algorithms, including empirical Bayesian Lasso (EBlasso) [53] and EN (EBEN) [54] based on penalized regression; eXtreme Gradient Boosting (XGBoost) [55] and Light Gradient Boosting Machine (LightGBM) [56] with Bayesian optimization, based on GBDT; and TabNet [59] and TabPFN [60], based on tabular deep learning, may improve prediction in the original dataset, as well as compare with other published models (Table 1).

The AD classification model developed with EBlasso included seven protein predictors and yielded a decent area under the ROC curve (AUROC) [95% confidence interval (CI)] in both the training and test sets (0.952 [0.905–1.000] and 0.971 [0.943–1.000], respectively) (Figure 1A,B; Table 2). The EBEN model included 9 proteins and also had a decent AUROC in both the training and test sets (0.966 [0.932–0.999] and 0.926 [0.874–0.978], respectively) (Figure 1C,D; Table 2). The XGBoost model yielded improved AUROC in the training set with comparable measures in the test set (0.999 [0.996–1.000] and 0.965 [0.922–1.000], respectively) (Figure 1E,F; Table 2). The LightGBM model yielded high AUROC in both training and test sets (1.000 [0.993–1.000] and 0.980 [0.955–1.000], respectively) (Figure 1G,H; Table 2), suggesting that GBDT may improve prediction. The TabNet model also had decent AUROC in both the training and test sets (0.935 [0.884–0.982] and 0.939 [0.876–1.000], respectively) (Figure 1I,J; Table 2), and the TabPFN model, which represents a cutting-edge approach to model development based on a pre-trained foundation model, also yielded high AUROC in both the training and test sets (1.000 [1.000–1.000] and 0.979 [0.956–1.000], respectively) (Figure 1K,L; Table 2).

Each of the models yielded relatively high accuracy, sensitivity/recall, specificity, positive predictive value (PPV)/precision, and negative predictive value (NPV), with the LightGBM model showing relatively the best test set metrics (Table 1). Therefore, despite using the same dataset, employing newer approaches to develop AD versus CN classification models, especially EBlasso, XGBoost, LightGBM, and TabPFN, showed improvement compared to the originally reported model using the older PAM algorithm [35] (Table 1). These models also showed similar or higher performance metrics compared to other previously published models, although it is not possible to make direct comparisons, given differences in the datasets across studies.

We evaluated the coefficient estimates for the EBlasso and EBEN models, or mean Shapley Additive Explanations (SHAP) values for XGBoost, LightGBM and TabPFN models, and feature importance for the TabNet model, to evaluate how each of the proteins included in various models contributes to the AD outcome prediction. The proteins selected in the EBlasso model included Angiopoietin-2 (ANG-2/ANGPT-2), Interleukin 1 alpha (IL-1α), Epidermal growth factor (EGF), Interleukin 3 (IL-3), Interleukin 11 (IL-11), Platelet-derived growth factor subunit B (PDGF-BB), and Tumor necrosis factor-alpha (TNF-α) (Figure 1B; Appendix A). The EBEN model additionally selected B lymphocyte chemoattractant (BLC) and Macrophage colony-stimulating factor (M-CSF/CSF1) (Figure 1B; Appendix A). There were 36 proteins with a mean SHAP value above 0, identified with XGBoost (Figure 1F; Appendix A), and 20 proteins with the LightGBM model (Figure 1H; Appendix A). The top ten proteins identified with the largest mean SHAP values in the XGBoost and LightGBM models also included ANG-2/ANGPT-2, IL-1α, IL-11, and PDGF-BB, as well as other interleukins, granulocyte-colony stimulating factor (G-CSF) and Monocyte chemotactic protein-3 (MCP-3/CCL7). The TabNet model included 26 proteins with feature importance above 0.01 (Figure 1J; Appendix A), including ANG-2/ANGPT-2, EGF, IL-1α, PDGF-BB, similar to other models, in addition to other proteins. The TabPFN model included 26 proteins with mean SHAP values above 0.01 (Figure 1L; Appendix A), which also included ANG-2/ANGPT-2, EGF, IL-1α, and PDGF-BB, among others.

### 2.2. Proteins Found to Be Important for Prediction Across Different Models Are Associated with Various Functions and Pathways

Overlaps in proteins identified as important predictors for AD across the six models were evaluated. There were four proteins, ANG-2/ANGPT-2, EGF, IL-1α, and PDGF-BB, included across all the models (Figure 2A). Furthermore, five of the models included IL-3, IL-11, M-CSF/CSF1, and TNF-α, four of the models included IL-1RA/IL1RN, and three of the models included CKβ-8-1/CCL23, G-CSF/CSF3, IL-1β, and I-TAC/CXCL11.

To better characterize the molecular pathways associated with these 13 commonly found proteins, we evaluated significantly enriched Gene Ontology (GO) Biological Processes (BP) (Figure 2B; Appendix A) and Kyoto Encyclopedia of Genes and Genomes (KEGG) pathway terms (Figure 2C; Appendix A). We found GO BP terms related to key cellular processes, including cell proliferation, division, and migration; terms related to relevant signaling pathways, including peptidyl-tyrosine phosphorylation, mitogen-activated protein kinase (MAPK), and cytokine signaling; as well as terms related to immune cell functions, including chemotaxis. KEGG pathway terms related to various signaling cascades related to cytokine and inflammatory responses, and cell proliferation and survival, including MAPK, phosphatidylinositol 3-kinase (PI3K)/protein kinase B (AKT), Janus kinase/signal transducer and activator of transcription (JAK/STAT), Ras-related protein 1 (Rap1), Ras and TNF signaling were identified, as well as terms related to cytokines, rheumatoid arthritis, and hematopoietic cell lineage. We also constructed a STRING protein–protein interaction network (Figure 2D).

We also performed a literature review to describe previous studies that have found these proteins, or genes encoding them or their transcripts, to be biomarkers associated with AD, as well as mechanistic studies in model organisms and cell culture that demonstrated their potential roles in AD development (Appendix A). Many studies were found, providing ample evidence demonstrating their roles as biomarkers, or their mechanistic involvement, such as by regulating the blood-brain barrier (BBB), cerebrovascular functions, microglia and astrocyte functions, inflammation, neuronal survival, and Aβ and Tau regulation.

### 2.3. The AD Biomarkers Identified Show Overrepresentation of Aging-Related Biomarkers

Given the known importance of aging as a major risk factor for AD, we evaluated how each biomarker identified may be related to aging by evaluating overlaps with previously published blood aging clock models or found to be differentially regulated by aging, based on the DNA methylome [62,63,64,65], transcriptome [66], and proteomics [67,68,69,70,71], or included in the Human Ageing Genomic Resources (HAGR) database (GenAge, CellAge, and cell senescence signatures) [72,73]. Consistent with the notion that aging is a major risk factor for AD, we found that the biomarkers were significantly overrepresented with known aging-related biomarkers and functions (*p* = 0.040) (Figure 3; Appendix A).

### 2.4. The Different Models Showed Variable Performance in Predicting Other Dementia or MCI Progression to AD

In addition to AD and CN plasma protein measures, the original publication [35] provided plasma protein measures of 11 subjects with other dementia (OD). The models developed earlier for AD versus CN classification were tested in this dataset for their potential ability to distinguish between AD and OD. The EBlasso and EBEN models predicted 90.9% of OD subjects correctly as non-AD (Table 3), similarly to the original study. The XGBoost and LightGBM models showed improvement, with 100% of OD subjects correctly predicted as non-AD (Table 3). The TabNet and TabPFN models performed much worse than the other models (Table 3).

Proteomic measures were available for MCI subjects, among whom *n* = 22 later developed AD (after a mean follow-up of 29.6 months), *n* = 1 developed FTD, *n* = 3 developed LBD, *n* = 4 developed VaD, and *n* = 17 remained as MCI (after a mean follow-up of 27.8 months). The models developed for AD versus CN classification were also tested in this dataset to evaluate their potential for detecting MCI to AD progression versus non-progression, or MCI to OD progression. The EBlasso and EBEN AD versus CN classification models predicted future MCI to AD progression for 86.4% or 90.9% of the subjects (*n* = 22), while predictions with the other models were lower (77.3% for XGBoost, 72.7% for LightGBM, 63.6% for TabNet, and 81.8% for TabPFN) (Table 4). All models correctly predicted MCI subjects who later developed FTD, LBD, or VaD (*n* = 8 total for all conditions) as non-AD. The models showed variability in predicting MCI subjects who remained as MCI after follow-up (*n* = 17), in which EBlasso, XGBoost and TabPFN predicted *n* = 4 as non-AD and *n* = 13 as non-AD, LightGBM predicted *n* = 7 as non-AD and *n* = 10 as AD, and EBEN and TabNet predicted *n* = 8 as non-AD and *n* = 9 as AD. These results may indicate that the heterogeneity in the expression levels of the proteins in the model during the MCI stage make it challenging to yield accurate predictions, or that a longer follow-up is needed to acquire more data on the patients’ future long-term outcomes to provide a better assessment of whether the models indeed distinguish between those who will develop AD and other outcomes and those who will never develop outcomes. Furthermore, since the models tested were initially trained to classify AD versus CN, and with the small sample sizes of all the different groups, additional studies to develop models specifically for the prediction of the progression of MCI to AD versus other outcomes are expected to be highly beneficial.

## 3. Discussion

The objective of this study was to apply various cutting-edge ML algorithms (EBlasso, EBEN, XGBoost, LightGBM, TabNet, and TabPFN) to a previously published plasma proteomics dataset [35] to develop AD outcome classification models and compare their performance to the original study, as well as with other studies. Each of the models we developed showed high AUROC, sensitivity/recall, specificity, PPV/precision, and NPV. Most of our models achieved improved accuracy compared to the model described in the original study, and given that the same dataset was used, our results support the notion that novel ML algorithms improve the development of AD plasma biomarkers. Our models were also comparable or showed better performance metrics compared to various other previously published plasma proteomic AD outcome prediction models; however, given that different datasets were used, it is not possible to make direct comparisons, as in the original study’s results. The training and test sets used to develop and assess the models in this study were relatively small, and the MCI or OD subgroup datasets were particularly small. To better evaluate whether the models are generalizable and not overfit, as well as the potential clinical utility, additional external validation studies with more patients are needed.

Our models identified various proteins to be important predictors for AD versus CN classification (7 proteins for EBlasso, 9 for EBEN, 36 for XGBoost, 20 for LightGBM, 26 for TabNet, and 26 for TabPFN), and further functional enrichment and literature review suggested that they may indeed be relevant to AD outcome. The models developed with EBlasso and EBEN yielded simple, regression-based prediction methods with only a handful of proteins, whereas the XGBoost, LightGBM, TabNet, and TabPFN models yielded more complex models that included more protein predictors. The protein predictor panels can be measured with a multiplex enzyme-linked immunosorbent assay (ELISA) or Luminex assays to facilitate implementation in the clinical setting.

We aimed to understand the underlying molecular mechanisms associated with the proteins conferring prediction by functional enrichment, as well as by literature review. ANG-2/ANGPT-2, EGF, IL-1α, and PDGF-BB were identified by all six algorithms, and each had ample prior studies linking them mechanistically to AD, providing strong evidence for their relevance (see Appendix A). Other proteins identified by at least three of the six models tested include IL-3, IL-11, M-CSF/CSF1, TNF-α, IL-1RA/IL1RN, CKβ-8-1/CCL23, G-CSF/CSF3, IL-1β, and I-TAC/CXCL11, which also had previous studies demonstrating their potential role in AD outcome (see Appendix A). Collectively, these proteins were associated with the enrichment of GO BP and KEGG pathway terms related to cytokines and the MAPK, PI3K/AKT, and JAK/STAT pathways, suggesting that these pathways may represent targets for developing novel preventative or therapeutic interventions. These findings are supported by previous works that described the importance of addressing the JAK/STAT pathway [25], aging-related neuroinflammation [26], and use of inhibitors against components of the pathways related to EGF [74], PDGF-BB [75], MAPK [75], or relevant cytokine receptors [76,77,78,79], as potentially ways to counter AD.

Ample previous studies have suggested that ANG-2/ANGPT-2 plays a role in AD pathology, especially through mediating angiogenesis and impacting the BBB. It is well-established that ANG-2 promotes angiogenesis through the TEK receptor tyrosine kinase (TIE-2) and inhibition of ANG-1 [80]. ANG-2 gain-of-function mice display increased BBB permeability, with downregulated tight junctions and adherens in endothelial cells, and upregulation of caveolin-1, which promotes permeability [81]. During murine development, Ang-2/Angpt-2, along with vascular endothelial growth factor (VEGF), promotes increased permeability and angiogenesis of the pupillary membrane micro-vessels [82]. In patients, ANG-2 was found to be upregulated in the CSF and correlated with the disease pathology, t-Tau, and p-Tau, as well as BBB permeability [83], showing its potential use as a biomarker. In the postmortem brains of AD patients, it was also upregulated, compared to controls [84], along with *TIE-2*, notably in endothelial cells, as well as altered in BA7 tissue homogenate expression, depending on Braak stages, as with TIE-2 [85], further showing that cerebrovasculature and BBB dysregulation due to ANG-2:TIE-2 signaling is mechanistically relevant. In a murine Amyloid Precursor Protein (APP) transgenic model, Aβ was found to upregulate Ang-2 in the cortex and hippocampus, leading to pathological angiogenesis [86]. In another human AβPP transgenic murine model, Ang-2 and Tie-2 were both shown to be upregulated [87]. In patients, ANG-2 has also been associated with various inflammatory conditions, including autoimmune diseases, sepsis, and acute lung injury [88]. Another study showed that in the presence of *Mycoplasma pulmonis* infection and thus high inflammation condition, Ang-2 binds Tie-2 in an antagonistic manner, suppressing its downstream phosphorylation and promoting forkhead box O1 (FoxO1) activation and increased Ang-2 expression by a positive feedback loop, leading to increased pathological vascularity and vessel permeability; whereas in the absence of the pathogen, Ang-2 activates Tie-2, leading to enlarged vessels without leakiness [89]. Collectively, these studies suggest the role of ANG-2 overexpression in pathological vasculature and BBB permeability to promote AD pathology.

EGF, which was also identified to be among the most relevant to AD classification, has also been shown extensively to be relevant to AD pathogenesis. Lower plasma EGF at baseline was shown to predict worse long-term cognitive outcomes in AD cohorts, and it was also found to be reduced among MCI and AD patients, compared to CN controls [90]. Other studies, on the other hand, have reported elevated plasma EGF in AD patients compared to controls [91,92]. A study in a murine model overexpressing human APOE4, with increased Aβ_1–42_ burden and cognitive impairment, lower plasma EGF was also reported, and EGF treatment was shown to alleviate cognitive decline, reduce microbleeds and increase cerebrovascular coverage [93,94]. Further supporting that EGF may be protective of AD, a study of single brain endothelial cell cultures and triple cultures (endothelial cells, astrocytes, and pericytes) mimicking a microvascular unit with oligomeric Aβ_1–42_ results in decreased angiogenesis and increased vessel disruption; however, EGF treatment prevented this effect, suggesting its use as a potential therapeutic for AD [95]. It is also well-established that the stimulation of its receptor EGFR by Aβ_1–42_, is one mechanism that promotes AD pathogenesis [96], and pharmacological inhibitors against EGFR have been shown to improve memory behavioral outcomes in *Drosophila* and murine models [74], as well as reduce Tau-induced neuroinflammatory response, microglia and astrocyte activation, and Tau hyperphosphorylation [97]. It is thus speculated that promoting EGF stimulation of EGFR, instead of by Aβ_1–42_, may aid in ameliorating AD outcomes.

Various cytokines were identified, including IL-1α by all six models; TNF-α, IL-3, and IL-11 by five of the models; and IL-1β by three models. IL-1α is a well-established pro-inflammatory cytokine which has been shown to be increased in the serum of AD patients, along with its family member IL-1β, and their antagonist IL-1RA and soluble receptor sIL-1R1 [98]. IL-1β and IL-RA were also both identified by at least three models. Plasma IL-1α was also found to be correlated with cognitive tests and Aβ40 [99]. IL-1 was also found to be elevated in postmortem brain samples from AD patients [100]. Polymorphisms of the *IL-1α* gene have also been associated with AD [101,102,103,104,105]. Moreover, in the acute phase of head injury, known to augment AD risk afterward, the number of IL-1α expressing activated microglia was found to be increased and correlated with the number of neurons with higher APP expression [106]. Mechanistically, in human astrocytes, IL-1α and IL-1β can lead to increased APP translation, and further assessment with a reporter system showed that they regulate the 5′-untranslated region (UTR) of APP [107]. Additionally, IL-1α was found to upregulate α-disintegrin and metalloproteinase (ADAM)-10 and -17, promoting soluble amyloid precursor protein-α (sAPPα) release [108]. It was further shown that this IL-1α stimulation of sAPPα secretion depended on initial p38 MAPK activation and subsequent MEK and PI3K activation, further elucidating the specific pathways involved [107]. In addition to its role in inflammation and APP processing, IL-1α induces the free radical nitric oxide in primary human astrocytes [109]. Other cytokines and chemokines identified include IL-3, IL-11, M-CSF/CSF1, TNF-α, CKβ-8-1/CCL23, G-CSF/CSF3, IL-1β, and I-TAC/CXCL11. Ample evidence links them to AD, including studies of plasma, serum, CSF biomarkers, and polymorphisms, and by mechanistic studies (see Appendix A). Taken together, there is ample literature to support the involvement of these biomarkers mechanistically in AD pathogenesis.

PDGF-BB was also identified as an important predictor across all six models. In a previous study, it was found to be elevated in AD patients’ plasma [91], CSF [110], and postmortem frontal cortex samples [111], compared to controls, as well as elevated in the CSF of MCI patients and in an AD murine model [112], compared to controls. It is well established that endothelial-derived, secreted PDGF-BB induces PDGF receptor-β (PDGFRβ) signaling in pericytes, which is crucial for the maintenance of the integrity of the blood-brain barrier [113,114]. Deficiency in pericytes and age-related vascular damage have been found to lead to neurodegeneration, neuroinflammation, and learning and memory loss [113]. Pericyte proliferation through PDGF-BB signaling has been described to be protective of AD-related neuronal damage in both murine models of Aβ pathology and human cells [74]. PDGF-BB has also been shown to confer protection against neuronal death from ischemic neuronal damage [115,116] and Parkinson’s disease [117,118]. Murine studies also support the mechanism related to aberrant PDGF-BB elevation in BBB permeability and neuroinflammation [119,120]. These studies support the notion that proper regulation of PDGF-BB is important for the maintenance of proper cerebrovasculature and BBB maintenance to prevent AD and cognitive decline.

Furthermore, we explored whether the biomarkers identified by the six models constructed were significantly enriched with previously established aging-related biomarkers [62,63,64,65,66,67,68,69,70,71,72,73]. We found a significant overrepresentation of aging-related biomarkers among the AD biomarkers in the models, corroborating the notion that aging is a major risk factor for AD. Our study characterized specific aging-related biomarkers, and additional studies to explore overlaps with biomarkers of other aging-related conditions and the possibility of developing anti-aging interventions to broadly prevent or delay aging-related diseases, are expected to be highly advantageous.

As a method for predicting AD outcome among MCI patients years before diagnosis, all the models were able to detect the majority of those who developed AD after follow-up, although the TabNet model performed notably worse. The models classified MCI patients who later developed FTD, LBD, or VaD as non-AD, although further studies with additional samples to be able to develop prediction models specifically for MCI to AD, versus MCI to OD, versus MCI to CN would be beneficial. Similar to the result of the model described in the original study [35], the models constructed in this study predicted many of the MCI patients who remained as MCI after follow-up, as AD. The limited accuracy in the prediction of MCI may indicate heterogeneity in the proteins during the MCI stage, or, since it is unknown whether the MCI patients in this study later developed AD or OD after the study’s follow-up, the assessment time may not have been sufficient. Additional prospective studies are needed to evaluate how well the models can detect future MCI to AD transition, early. Furthermore, since the models were initially developed for the classification of AD versus CN, additional prospective studies to perform prediction model development among MCI patients confirmed to develop AD, versus those who never develop MCI or develop OD in the long run, would be beneficial. A major challenge with such a study is having sufficient follow-up time to be certain about the patients’ long-term outcomes, with sufficient sample sizes. The sample sizes for these subgroup assessments were particularly small, making it difficult to evaluate the clinical utility of this approach, and thus, future studies are required.

Overall, our results support the notion that modern ML algorithms can be powerful tools for developing classification models for AD, based on plasma protein signatures. Moreover, these algorithms allow us to distinguish AD from OD. Applying these analysis methods to datasets with recent methods for large proteome profiling that allow measuring many more proteins, with recent AD clinical classification, is expected to further improve disease detection. A major limitation of our study is that the dataset was derived from a previously published study without linked demographic or other clinical information, and in a relatively small population. Thus, a future study to validate our models using a large study base is necessary. The utilized dataset had limited MCI subgroup data, impeding the capacity of ML algorithms to discern the MCI subgroup transitioning from MCI to AD and OD. The ability to predict the progression of MCI patients to AD or OD holds significant clinical utility for healthcare providers [19,40]. It would also be advantageous to validate the biomarkers in various settings, including in targeted at-risk patient groups, or widely in the community setting, to evaluate whether they can be beneficial for screening.

Further limitations of this study are that clinical data associated with the proteomic data were not available, the criteria for AD diagnosis and cognitive norm were evaluated according to the labels provided by the original study only, information about key factors such as the subjects’ APOE ε4 genotype could not be accounted for, and measurements of Aβ and Tau peptides were not available to be able to make comparisons. Future large external validations in different populations are required to evaluate the generalizability and better assessment of potential overfitting of the models, with fully linked clinical and molecular information. In this study, the standard threshold of 0.5 was used to predict AD outcome, rather than employing threshold optimization, such as Youden’s Index, as this was a small study with equal cases to non-cases, and the sensitivity vs. specificity measures were comparable. With a future large study, it would be advantageous to evaluate how different thresholds that maximize both sensitivity and specificity, or prioritize sensitivity or specificity, may be advantageous in the clinical setting. Although more patients may be detected by prioritizing sensitivity, it may also be disadvantageous, given the additional cost imposed for diagnostic evaluation of those found to be at high risk and worry for those who may be false positives. Furthermore, future studies to evaluate whether employing simpler measures, such as p-Tau217/Aβ_1–42_ ratio is more cost-effective, or whether employing proteomic information may aid in predicting response to specific therapeutics, would be highly informative.

In the future, if these models can be validated, a rapid assay can be developed to facilitate measuring the biomarkers in the clinical setting. Moreover, molecular characterization to elucidate how the proteins identified may be related to AD pathogenesis, and exploring whether they are drivers of the disease status, may aid in the development of novel therapeutic strategies.

## 4. Materials and Methods

### 4.1. Dataset Used

This study used de-identified datasets originally provided as Appendix A by Ray et al. [35] and was determined not to meet the criteria for human subject research by the Mass General Brigham Institutional Review Board. The datasets entailed standardized measures of 120 plasma proteins assessed with filter-based, arrayed sandwich ELISAs [121]. The samples were derived from patients who were diagnosed with Alzheimer’s disease (AD) at the time of plasma collection or matched cognitively normal (CN) controls. According to the original study, the samples from patients diagnosed with clinical symptoms by neurologists were archived at various academic centers specializing in neurological or neurodegenerative diseases (Sun Health Research Institute, Oregon Health Sciences University, UC San Diego, University of Genoa, Göteborg University, University of Wroclaw, UC San Francisco, Stanford University). Although the final clinical labeling information was provided, linked clinical datasets with additional information and *APOE* genotype information were not available. The original study provided data from 83 subjects as the training set (*n* = 43 AD and *n* = 40 CN) and 81 as the test set (*n* = 42 AD and *n* = 39 CN). Additional data provided in the original study, which were also analyzed in this study, included the following: (i) subjects diagnosed with other dementia (OD) (*n* = 11 OD total, with *n* = 8 Frontotemporal dementia (FTD) and *n* = 3 Corticobasal degeneration (CBD)); (ii) subjects diagnosed with mild cognitive impairment (MCI) at the time of plasma collection, who developed AD after 2–5 years of follow-up (*n* = 22 MCI-AD); (iii) subjects diagnosed with MCI at the time of plasma collection, who developed OD (*n* = 8 MCI-OD total, with *n* = 1 MCI-FTD, *n* = 3 MCI-Lewy Body dementia (LBD), and *n* = 3 MCI-Vascular dementia (VaD)); and (iv) subjects diagnosed with MCI at the time of plasma collection, who remained as MCI patients after 4–6 years of follow up (*n* = 17 MCI-MCI).

### 4.2. Software

Python version 3.12.3 [122] and R version 4.4.2 [123] were used for the analyses described below. The code scripts and output are provided in a Github repository (https://github.com/additbio/AD-plasma-inflammatory-biomarkers (accessed on 18 October 2025)).

### 4.3. Development of the EBlasso, EBEN, XGBoost, LightGBM, TabNet, and TabPFN Models

EBlasso and EBEN were performed in the training set for feature selection using the “EBglmnet” R package version 6.0 [124], where 5-fold CV was performed to identify the hyperparameters that maximize the penalized likelihood function (for EBlasso, α = 1 and λ = 0.1703, and for EBEN, α = 0.300 and λ = 0.118). With the selected proteins (7 for EBlasso and 9 for EBEN), final models were constructed by performing multivariable logistic regression on the training set to estimate the maximum likelihood coefficient estimates with base functions. The coefficient estimates and 95% CI were plotted with the “ggplot2” R package version 3.4.4.

The XGBoost [55] model was constructed using the “xgboost” library version 2.1.1 [125], and the LightGBM [56] model was constructed using the “lightbgm” Python library version 4.5.0 [126]. Using the training set, Bayesian optimization was performed with 5-fold CV to determine the hyperparameter values that maximized the mean test AUROC, using the “scikit-learn” Python library version 1.5.2 [127] and “bayesian-optimization” Python library version 2.0.3 [128] (for XGBoost, learning rate = 0.1215, maximum depth of a tree = 4, gamma = 0.0526, minimum child weight = 3.2028, subsample ratio = 0.8090, column sample ratio = 0.9578, L1 regularization = 0.4681, L2 regularization = 2.7769, number of boosting rounds = 96; and for LightGBM, learning rate = 0.2653, maximum depth of a tree = 5, number of leaves in a tree = 16, minimum data in a leaf = 24, minimum sum of Hessian in a leaf = 0.001, subsample ratio = 0.8262, feature fraction = 0.9200, minimum gain to split = 1.0782, L1 regularization = 0.1931, L2 regularization = 1.3596, number of boosting rounds = 69). For the XGBoost and LightGBM models, SHapely Additive exPlanations (SHAP) scores [57,58] were found for each protein with the “shap” Python library version 0.46.0 [129]. The XGBoost model included 36 proteins, and the LightGBM model included 20 proteins with a mean absolute SHAP score above 0.

The TabNet model [59] was constructed using the “pytorch_tabnet” library version 4.1.0 [130]. Five-fold CV was performed to tune the hyperparameters that maximized the AUROC with the optimizer in the “pytorch” library version 2.3.1 [131] (mask type=“sparsemax”, width of decision prediction layer = 12, width of attention embedding = 12, number of steps = 4, gamma = 1.3, epsilon = 3.827 × 10^−15^, momentum = 0.028, sparsity loss coefficient = 8.866 × 10^−5^, AMGrad optimizer with initial learning rate = 0.008 and weight decay = 1 × 10^−5^, Reduce on Plateau learning rate scheduler, with minimum learning rate = 0.001, factor = 0.1, patience = 10, batch size = 16, virtual batch size = 8, epochs = 109, seed = 396). The top feature importance was plotted with the “ggplot2” R package version 3.4.4. The TabPFN model [60,61], which does not require hyperparameter tuning, was constructed with the “tabpfn” library version 2.0.6 [132] and SHAP scores were obtained with the “tabpfn_extensions” library [133] version 0.0.4 in Python.

Each final model constructed in the training set was applied to the test set, and predicted probabilities were obtained. The AUROC [95% CI] was calculated and curves were plotted with the “pROC” R package version 1.18.5 [134]. The sensitivity, specificity, PPV, and NPV, and their 95% CIs were calculated using the “epiR” R package version 2.0.78 [135]. A predicted probability threshold cut-off of 0.5 was used when determining AD outcome prediction.

### 4.4. GO and KEGG Pathway Enrichment and Network Analysis of the Overlapping Proteins

A Venn diagram was plotted to display the proteins commonly found across all the models, using the “ggVennDiagram” R package version 1.5.4 [136]. Using the 13 proteins found in at least three models, functional annotation enrichment analysis for Gene Ontology (GO), Biological Process (BP), and Kyoto Encyclopedia of Genes and Genomes (KEGG) pathway terms was performed with minimum signal and strength settings ≥1.00, and a Protein–Protein Interaction (PPI) Network plot was constructed with the high confidence (0.700) interaction score setting based on curated databases, experimental evidence, text mining, and co-expression (as indicated in the figure), using STRING version 12.0 [137,138].

### 4.5. Evaluating Overlaps with Aging Models Developed in Blood

Proteins identified to be important for AD prediction were compared with previously published DNA methylome-based [62,63,64,65], transcriptome-based [66], and protein-based [67,68,69,70,71] aging clock models, and previously known aging-related proteins described in the publications [68,70]. The Human Ageing Genomic Resources (HAGR) database (GenAge, CellAge, and cell senescence signatures) [72,73] was also used to identify aging-related gene encodings for relevant proteins. The ENTREZ identification numbers provided in the original AD dataset [35] were used to make comparisons. For studies where an ENTREZ identification number was not provided, the “AnnotationDbi” R package version 1.68.0 [139] with “org.Hs.eg.db” version 3.20.0 [140] were used to find the associated numbers. A hypergeometric test *p*-value was calculated with the R base stats Fisher’s exact test function, with an alternative hypothesis of greater than to assess overrepresentation of aging-related biomarkers.

## Figures and Tables

**Figure 1 ijms-26-11673-f001:**
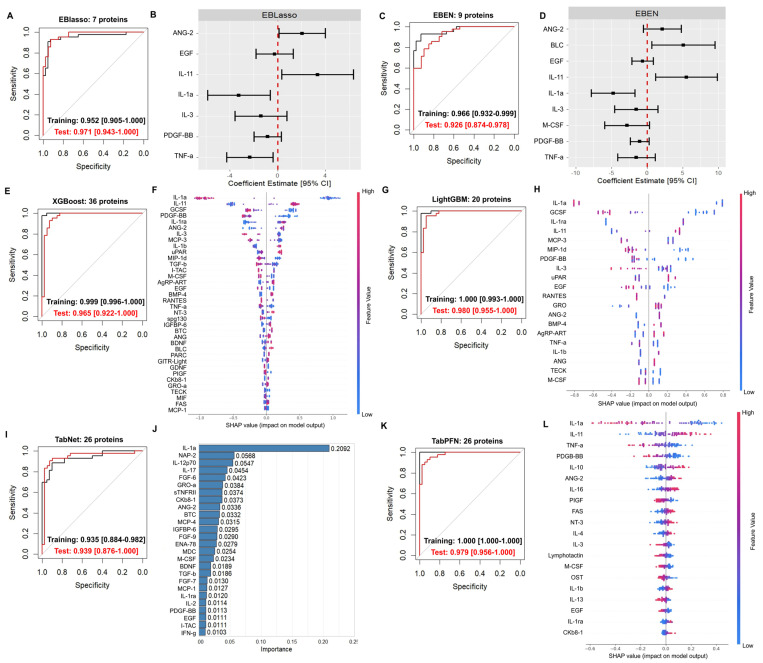
ROC curves and AUROC [95%CI] of the models developed with (**A**) EBlasso showing (**B**) coefficient estimates, (**C**) EBEN with (**D**) coefficient estimates, (**E**) XGBoost with (**F**) mean SHAP scores, (**G**) LightGBM with (**H**) mean SHAP scores, (**I**) TabNet with (**J**) top feature importance scores above 0.01, and (**K**) TabPFN with (**L**) top mean SHAP scores above 0.01.

**Figure 2 ijms-26-11673-f002:**
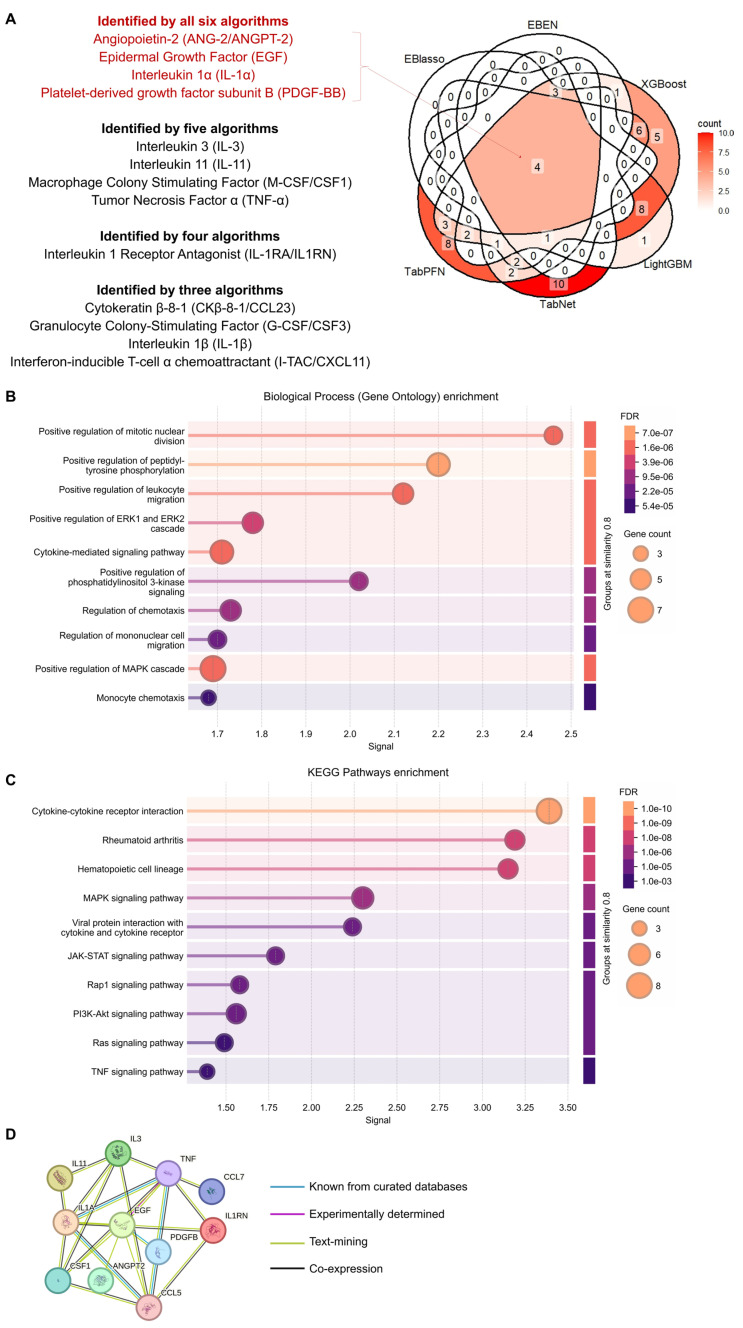
Overlaps across the predictive proteins included in the different models developed, and functional assessment. (**A**) Venn diagram showing overlaps in predictive proteins included in each of the classification models developed. The numbers show the total number of overlapping proteins, and the colors correspond to the number, with red being a higher count, as shown in the legend. (**B**) Gene ontology and (**C**) KEGG pathway enrichment analysis result of the 13 proteins identified by at least three of the six models developed, with (**D**) STRING PPI network analysis.

**Figure 3 ijms-26-11673-f003:**
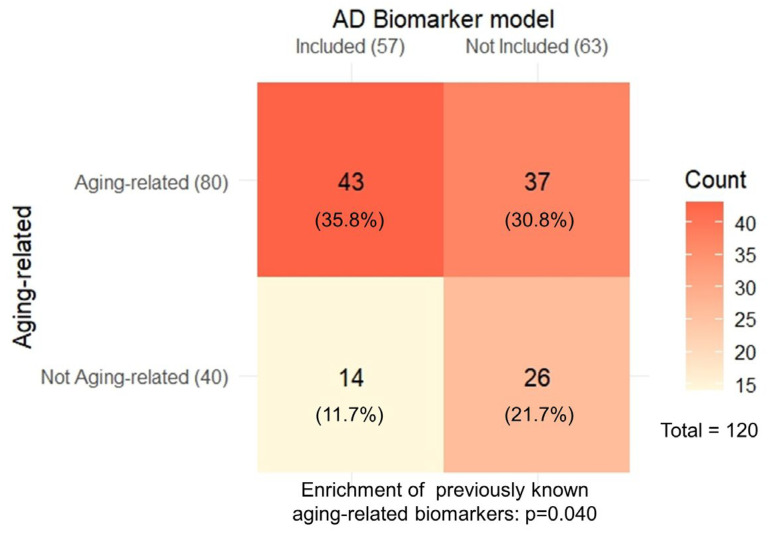
Assessment of the relevance of underlying molecular mechanisms of aging, where the AD biomarkers identified by the ML models were significantly enriched with previously described aging-related biomarkers.

**Table 1 ijms-26-11673-t001:** Previously reported plasma or serum molecular biomarker classification or prognostic models aside from those based only on Aβ and Tau peptides. The original study from which the dataset for this study was derived (Ray S. et al., 2007 [35]) is indicated in bold.

AD vs. CN Classification Models		
PAM ML 18 Plasma Proteins	Accuracy = 89% (Training and Test)	Ray S. et al., 2007 [35]
SVM ML 7 to 10 plasma proteins	AUC = 0.86 to 0.89	Eke CS. et al., 2021 [37]
Four models with 5 to 14 plasma proteins	AUC = 0.759 to 0.838 (CV), 0.737 to 0.842 (ext. valid.)	Llano DA. et al., 2013 [45]
9 plasma proteins	AUC = 0.79 (training and ext. valid.)	Sung YJ. et al., 2023 [34]
5 plasma proteins + age + *APOE* genotype	AUC = 0.79, 0.81 (ext. valid.)	Morgan AR. et al., 2019 [46]
Ridge + SVM ML 11 plasma proteins + age	AUC = 0.891 (test)	Ashton NJ. et al., 2019 [38]
19 hub plasma proteins	AUC = 0.969 (ext. valid.)	Jiang Y. et al., 2022 [47]
SVM ML 4 plasma + 6 serum proteins	AUC = 99.98% (training), 93.96% (test)	Zhang F. et al., 2022 [39]
LGBM ML 4 plasma proteins + demographic + cognition	AUC = 0.913	Guo Y. et al., 2024 [41]
Lasso ML 7 plasma proteins	AUC = 0.796 (test), 0.721 (replication), 0.715 and 0.757 (ext. valid.)	Heo G. et al., 2025 [42]
Random Forest ML 14 serum proteins	AUC = 0.91 (training), 0.88 (test)	O’Bryant SE. et al., 2010 [36]
21 serum proteins + age + sex + education	AUC = 0.89	O’Bryant SE. et al., 2016 [44]
XGBoost ML plasma metabolite	AUC = 0.88 (test)	Stamate D. et al., 2019 [49]
12 serum miRNA	Accuracy = 76%	Zhao X. et al., 2020 [48]
**AD vs. MCI Classification or Prognostic Models**		
3 plasma proteins	AUC = 0.74 (training), 0.67 (ext. valid.)	Morgan AR. et. al., 2019 [46]
SVM ML 7 to 10 plasma proteins	AUC = 0.80 to 0.83	Eke CS. et. al., 2021 [37]
Lasso-ML selected 12 plasma proteins + plasma Aβ + plasma pTau + baseline cognitive measures + age + sex + education + *APOE* genotype	AUC = 0.88, accuracy = 86.7% (test)—prognostic model for MCI-progressors vs. MCI-stable	Kivisäkk P. et. al., 2022 [40]

**Table 2 ijms-26-11673-t002:** Sensitivity/recall, specificity, Positive Predictive Value (PPV)/precision, and Negative Predictive Value (NPV) with 95% CI of each of the prediction models for AD vs. CN.

	EBlasso (7 Proteins)	EBEN (9 Proteins)	XGBoost (36 Proteins)	LightGBM (20 Proteins)	TabNet (27 Proteins)	TabPFN (26 Proteins)
**Training set**						
**Accuracy**	0.916 [0.834–0.965]	0.916 [0.834–0.965]	0.988 [0.935–1.00]	0.964 [0.898–0.992]	0.831 [0.733–0.905]	0.988 [0.935–1.000]
**Sensitivity/recall**	0.930 [0.809–0.985]	0.907 [0.779–0.974]	0.977 [0.877–0.999]	0.977 [0.877–0.999]	0.884 [0.749–0.961]	1.00 [0.918–1.000]
**Specificity**	0.900 [0.763–0.972]	0.925 [0.796–0.984]	1.000 [0.912–1.000]	0.950 [0.831–0.994]	0.775 [0.615–0.892]	0.975 [0.868–0.999]
**PPV/precision**	0.909 [0.783–0.975]	0.929 [0.805–0.985]	1.000 [0.916–1.000]	0.955 [0.845–0.994]	0.809 [0.667–0.909]	0.977 [0.880–0.999]
**NPV**	0.923 [0.791–0.984]	0.902 [0.769–0.973]	0.976 [0.871–0.999]	0.974 [0.865–0.999]	0.861 [0.705–0.952]	1.00 [0.910–1.000]
**Test set**						
**Accuracy**	0.914 [0.830–0.965]	0.827 [0.727–0.902]	0.926 [0.846–0.972]	0.938 [0.862–0.980]	0.901 [0.815–0.956]	0.926 [0.846–0.972]
**Sensitivity/recall**	0.929 [0.805–0.985]	0.833 [0.686–0.930]	0.929 [0.805–0.985]	0.952 [0.838–0.994]	0.929 [0.805–0.985]	0.929 [0.805–0.985]
**Specificity**	0.897 [0.758–0.971]	0.821 [0.665–0.925]	0.923 [0.791–0.984]	0.923 [0.791–0.984]	0.872 [0.726–0.957]	0.923 [0.791–0.984]
**PPV/precision**	0.907 [0.779–0.974]	0.833 [0.686–0.930]	0.929 [0.805–0.985]	0.930 [0.809–0.985]	0.886 [0.754–0.962]	0.929 [0.805–0.985]
**NPV**	0.921 [0.786–0.983]	0.821 [0.665–0.925]	0.923 [0.791–0.984]	0.947 [0.823–0.994]	0.919 [0.781–0.983]	0.923 [0.791–0.984]

**Table 3 ijms-26-11673-t003:** Application of the different prediction models to samples labeled as other dementia (OD), and evaluating the proportion correctly classified as not AD.

Correct Classification	EBlasso (7 Proteins)	EBEN (9 Proteins)	XGBoost (36 Proteins)	LightGBM (20 Proteins)	TabNet (26 Proteins)	TabPFN (26 Proteins)
**Other dementia** **(*n* = 11)**	10 (90.9%)	10 (90.9%)	11 (100%)	11 (100%)	7 (63.6%)	9 (81.8%)

**Table 4 ijms-26-11673-t004:** Application of the various prediction models to samples from MCI patients (*n* = 46) who later developed AD (after a mean follow up of 29.6 months), or who developed OD (Frontotemporal Dementia (FTD) or Lewy Body Dementia (LBD) or Vascular Dementia (VaD)) or remained as MCI (after a mean follow up of 27.8 months). For MCI-AD or MCI-OD outcomes, the number of correct predictions (%) is shown, and for MCI-MCI, the number of not AD/AD predictions is shown.

Correct Classification	EBlasso (7 Proteins)	EBEN (9 Proteins)	XGBoost (36 Proteins)	LightGBM (20 Proteins)	TabNet (26 Proteins)	TabPFN (26 Proteins)
**MCI-AD** **(*n* = 22)**	19 (86.4%)	20 (90.9%)	17 (77.3%)	16 (72.7%)	14 (63.6%)	18 (81.8%)
**MCI-OD-FTD** **(*n* = 1)**	1 (100%)	1 (100%)	1 (100%)	1 (100%)	1 (100%)	1 (100%)
**MCI-OD-LBD** **(*n* = 3)**	3 (100%)	3 (100%)	3 (100%)	3 (100%)	3 (100%)	3 (100%)
**MCI-OD-VaD** **(*n* = 4)**	4 (100%)	4 (100%)	4 (100%)	4 (100%)	4 (100%)	4 (100%)
**MCI-MCI** **(*n* = 17)**	4 not AD/ 13 AD	8 not AD/ 9 AD	4 not AD/ 13 AD	7 not AD/ 10 AD	8 not AD/ 9 AD	4 not AD/ 13 AD

## Data Availability

Restrictions apply to the availability of these data. Data were obtained from Ray et al. [35] Appendix A, and the link for their access and code for preprocessing them for this study are available at https://github.com/additbio/AD-plasma-inflammatory-biomarkers with the permission of Ray et al. [35].

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
