# Peer review of "Development of Plasma Protein Classification Models for Alzheimer’s Disease Using Multiple Machine Learning Approaches"

_ijms, 2025, doi:10.3390/ijms262311673_

Round 1

Reviewer 1 Report

Comments and Suggestions for Authors

I would like to thank the authors for this research under the title: "Development of plasma protein classification models for Alzheimer’s Disease using multiple machine learning approaches".

The research used established algorithms; EBlasso, EBEN, XGBoost, LightGBM, TabNet and TabPFN, in order to reach a conclusion that biomarkers in the plasma (Angiopoietin-2 (ANG-2), Interleukin 1α (IL-1α), and Platelet-derived growth factor subunit B (PDGF-B) can be used for detection of Alzheimer's Disease. He concluded that these proteins can be a therapeutic targets and preventative for AD. The title and abstract are well written and references are sufficient. Meanwhile, I have the following comments:

Although the idea is new and promising, the scope of the research is out of the scope of the journal. As a follower for the articles in the International Journal of Molecular Sciences (IJMS), which provides an important advanced forum (Q1 in Biochemistry and molecular biology and organic chemistry categories) for biochemistry, molecular and cell biology, molecular biophysics, molecular medicine, and all aspects of molecular research in chemistry. It is about mathematical models.

From my experience, it is suitable for journals in mathematics more than biology.

The reader of IJMS is seeking for biological information related to biochemistry more.

Author Response

The research used established algorithms; EBlasso, EBEN, XGBoost, LightGBM, TabNet and TabPFN, in order to reach a conclusion that biomarkers in the plasma (Angiopoietin-2 (ANG-2), Interleukin 1α (IL-1α), and Platelet-derived growth factor subunit B (PDGF-B) can be used for detection of Alzheimer's Disease. He concluded that these proteins can be a therapeutic targets and preventative for AD. The title and abstract are well written and references are sufficient.

We thank Reviewer 1 for recognizing the strength of our study.

Meanwhile, I have the following comments:

Although the idea is new and promising, the scope of the research is out of the scope of the journal. As a follower for the articles in the International Journal of Molecular Sciences (IJMS), which provides an important advanced forum (Q1 in Biochemistry and molecular biology and organic chemistry categories) for biochemistry, molecular and cell biology, molecular biophysics, molecular medicine, and all aspects of molecular research in chemistry. It is about mathematical models.

From my experience, it is suitable for journals in mathematics more than biology.

The reader of IJMS is seeking for biological information related to biochemistry more.

We thank Reviewer 1 for pointing this out. To improve the relevance of our study, we added an extensive literature review (Supplementary Table S7, p7-11 in Supplementary Materials) describing known molecular mechanisms related to AD for the biomarkers uncovered. We describe having performed this additional literature review on p.8 of the main manuscript, as well as describe the mechanisms in the discussion section, p.10-12 of the main manuscript. We hope that the addition of this literature review would help make the content of our study more relevant to the readership.

Reviewer 2 Report

Comments and Suggestions for Authors

This is a well-motivated and technically sound manuscript that explores the use of various cutting-edge machine learning (ML) algorithms for plasma proteomic classification of Alzheimer’s Disease (AD). The authors make strong use of a legacy dataset and modern interpretability tools (e.g., SHAP, TabNet, TabPFN) to enhance model performance and provide biological insights. However, the paper merits publication after revisions addressing the following concerns:

  • The dataset is relatively small (n=83 training, n=81 test), with even fewer samples in subgroups (e.g., n=11 other dementia). This raises concerns about overfitting, especially given the high AUROC values (>0.96) across all models.
  • The use of SHAP values, feature importance plots, and functional enrichment analysis provides biological interpretability of ML findings. Despite this, the reproducibility of the models is limited due to insufficient reporting of model hyperparameters, random seeds, and lack of open-source code or notebooks.
  • Discuss limitations in translating high AUROC models into clinically deployable tests. Include considerations for threshold tuning, false positives, and cost-effectiveness.

Author Response

This is a well-motivated and technically sound manuscript that explores the use of various cutting-edge machine learning (ML) algorithms for plasma proteomic classification of Alzheimer’s Disease (AD). The authors make strong use of a legacy dataset and modern interpretability tools (e.g., SHAP, TabNet, TabPFN) to enhance model performance and provide biological insights.

We are grateful to Reviewer 2 for recognizing the motivation for our study and its strengths.

However, the paper merits publication after revisions addressing the following concerns:

  • The dataset is relatively small (n=83 training, n=81 test), with even fewer samples in subgroups (e.g., n=11 other dementia). This raises concerns about overfitting, especially given the high AUROC values (>0.96) across all models.

We thank Reviewer 2 and agree with this concern, and accordingly, have made the following modifications: 1. Described this caveat and need for larger external validation studies in the abstract; 2. Toned down on describing the AUROC as exceptional in the results section, and to instead mention that using the same dataset, we achieved relatively better performance compared to the original study (p. 4-5); 4. More extensively described this caveat in the Discussions section (p. 10, 13); 5. Included a literature review mentioning previous studies that have shown mechanistic relevance of the biomarkers identified, or shown them to also be effective biomarkers to provide further evidence for the relevance of the biomarker model (Supplementary Table S7 in p. 7-12 of the Supplementary Materials section, Discussion in p. 10-12 main text). We hope that meticulously discussing the small dataset used, the possibility of overfitting, and the need for future external validation studies, will allow readers to better interpret the results and alleviate the concerns of the Reviewer.

  • The use of SHAP values, feature importance plots, and functional enrichment analysis provides biological interpretability of ML findings. Despite this, the reproducibility of the models is limited due to insufficient reporting of model hyperparameters, random seeds, and lack of open-source code or notebooks.

We appreciate Reviewer 2's suggestions, and accordingly, have created a Github repository with the R code provided in a R Markdown file with accompanying Renv .json file for reproducibility of package versions, and Python code provided in a Jupyter notebook with accompanying requirements file for library versions, as well as .html files that show the final output, and Readme that describes how to use the scripts on the original dataset. We provided the link to this Github page in the methods section (p.14 - this link is currently set to private, but will be made public upon publication). We have also added more details to the hyperparameters and random seed used in the methods section (p. 14-15). We are grateful to Reviewer 2 for this suggestion to strengthen our manuscript.

  • Discuss limitations in translating high AUROC models into clinically deployable tests. Include considerations for threshold tuning, false positives, and cost-effectiveness.

We thank Reviewer 2 for these thoughtful suggestions, and have included additional discussions about the caution interpreting high AUROCs and considerations of threshold optimization, potential drawbacks of prioritizing sensitivity versus specificity, and impact on cost-effectiveness in the Discussions section (p. 13). 

Reviewer 3 Report

Comments and Suggestions for Authors

The research aims to develop and compare diagnostic algorithms based on machine learning for the classification of Alzheimer's disease (AD) using plasma proteins. The use of plasma markers instead of cerebrospinal fluid (CSF) is a well-researched area. Therefore, the novelty lies not so much in finding biomarkers, but rather in the new diagnostic approach. Despite promising results, the paper contains several significant structural and methodological issues.

1. The text still includes instructions for authors (lines 41-49, 312-315, and 455-470) that should be removed. This suggests a lack of careful attention to the details of the manuscript.

2. The introduction should include the elements that are typically found in the "Discussion" section, such as an analysis of the algorithms used in previous studies and a preliminary discussion of the advantages of the author's new methods. The introduction should also set the context for the study by describing the problem that the research aims to address and the goals of the research. It should not include any discussion of specific results or conclusions until they are presented in the appropriate section.

3. Regarding the methodological limitations and lack of description in the Materials and Methods section, it is essential to provide a clear description of the criteria used for diagnosing Alzheimer disease. This information is crucial for understanding the validity of the data used in the study. In the section "Materials and methods" it is necessary to add a detailed description of the clinical criteria for the diagnosis of AD and cognitive norm in patients in the initial data set.

4. The APOE e4 allele is the strongest genetic risk factor for AD, and the authors do not explain why it was not taken into account in their models. This absence raises questions about the completeness of their models and their potential clinical applicability.

5. All models were tested using data from a single source, which is a significant limitation. The lack of external validation raises concerns about the generalizability of the results and the potential for clinical application. Claims of clinical applicability must be approached with caution, given the need for further validation in larger, independent cohorts.

Author Response

The research aims to develop and compare diagnostic algorithms based on machine learning for the classification of Alzheimer's disease (AD) using plasma proteins. The use of plasma markers instead of cerebrospinal fluid (CSF) is a well-researched area. Therefore, the novelty lies not so much in finding biomarkers, but rather in the new diagnostic approach.

We thank Reviewer 3 for this thoughtful response to our study's motivations and findings.

Despite promising results, the paper contains several significant structural and methodological issues.

  1. The text still includes instructions for authors (lines 41-49, 312-315, and 455-470) that should be removed. This suggests a lack of careful attention to the details of the manuscript.

We apologize for this error and have verified that the instructions have been removed in this revised version.

  1. The introduction should include the elements that are typically found in the "Discussion" section, such as an analysis of the algorithms used in previous studies and a preliminary discussion of the advantages of the author's new methods. The introduction should also set the context for the study by describing the problem that the research aims to address and the goals of the research. It should not include any discussion of specific results or conclusions until they are presented in the appropriate section.

We are grateful to Reviewer 3 for this suggestion, and have provided information about the advantages of each of the algorithms used in this study compared to the previous study, and the motivation for performing this study, which tests new analytic approaches that have not yet been used previously, and have also removed mention of specific results and conclusions from the Introduction section (p. 2-4).

  1. Regarding the methodological limitations and lack of description in the Materials and Methods section, it is essential to provide a clear description of the criteria used for diagnosing Alzheimer disease. This information is crucial for understanding the validity of the data used in the study. In the section "Materials and methods" it is necessary to add a detailed description of the clinical criteria for the diagnosis of AD and cognitive norm in patients in the initial data set.

We agree with Reviewer 3 - given that we are using a previously described dataset, we do not have further information, beyond what was provided by the original study. In the Materials and Methods section, we have described the dataset as much as possible based on the information provided in the original study, and have added a mention that further clinical information is not available (p. 14). We have also included a mention of this caveat in the Discussions section, and highlighted the need for an updated study with further clinical information (p. 13).

  1. The APOE e4 allele is the strongest genetic risk factor for AD, and the authors do not explain why it was not taken into account in their models. This absence raises questions about the completeness of their models and their potential clinical applicability.

We agree with Reviewer 3 - similar to the point made above, we unfortunately do not have further information, beyond what was provided by the original study. Therefore, we have also added a mention in the Materials and Methods section that APOE genotype information was not available (p. 14), as well as discussed that this information would have been useful, in the Discussions section (p. 13).

  1. All models were tested using data from a single source, which is a significant limitation. The lack of external validation raises concerns about the generalizability of the results and the potential for clinical application. Claims of clinical applicability must be approached with caution, given the need for further validation in larger, independent cohorts.

We thank Reviewer 3 for highlighting this point, and have more extensively discussed the need for further external validation in larger studies in the Abstract and Discussions section (p. 13). We hope that these additional explicit mentions and discussions of this limitation and making interpretations cautiously, would help alleviate your concerns.

Thank you for your time and helpful suggestions.

Round 2

Reviewer 1 Report

Comments and Suggestions for Authors

There is an added molecular mechanism parts and pathways to the old version. So, from my point of view, it is now falls within the scope of IJMSjournal. The manuscript is now suitable for the readers and increases the knowledge about the genetic pathways adding new dimentions for the algorithmic model for the interaction between proteins.

Reviewer 3 Report

Comments and Suggestions for Authors

The authors responded to all my comments.